# Thermal Deformation of PA66/Carbon Powder Composite Made with Fused Deposition Modeling

**DOI:** 10.3390/ma13030519

**Published:** 2020-01-22

**Authors:** Fei Li, Jingyu Sun, Hualong Xie, Kun Yang, Xiaofei Zhao

**Affiliations:** 1School of Information Science and Engineering, Shenyang University of Technology, Shenyang 110870, China; lifeisut@163.com; 2Department of Mechanical Engineering and Automation, Northeastern University, Shenyang 110819, China; hlxie@mail.neu.edu.cn (H.X.); yk1873639934@163.com (K.Y.); zhaoxiaofei2019@163.com (X.Z.)

**Keywords:** CPRP, material modeling, processing parameters, warp, max thermal deformation

## Abstract

Polyamide 66 (PA66) is a material with high wear resistance, toughness, and heat resistance. However, low stiffness and thermal deformation during thermal processes define applications in many conditions. Carbon powder efficiently enhances stiffness and reduces thermal deformation, which makes up defects of plastic materials. However, forming a composite with fused deposition modeling (FDM) that accumulates material to a specified location by melting plastic filaments is limited, including fluidity and viscosity to form normally. In this paper, filaments of polyamide 66 (PA66) reinforced with carbon powder were produced. Digimat was used to analyze the composite material properties of different carbon contents and predict the proper carbon content. Then, the material properties were imported to ANSYS software to simulate the thermal deformation of the workpieces during processing. It was verified that adding carbon powder is helpful in decreasing thermal deformation. Comparing experiments and simulations, we found that 20% carbon mass fraction was best, and that thermal deformation was minimal at 240 °C nozzle temperature while hot bed temperature was 90 °C. The optimal ratio of extrusion speed to filling speed was 0.87, and the best aspect ratio was 0.25.

## 1. Introduction

Fused deposition modeling (FDM) is a type of additive manufacturing that accumulates material to a specified location by melting plastic filaments. It commonly uses acrylonitrile butadiene styrene (ABS), polylactic acid (PLA), and poly-ether-ether-ketone (PEEK) [1]. It is applied in many fields because the available materials have poor hardness, heat and wear resistance, low mechanical strength, and undergo large deformation during temperature-changing processes [2]. To make up for these defects, carbon powder is added into the plastic materials. However, the carbon powder is crisp and hard. For FDM processing, the plastic needs certain viscosity, fluidity, and toughness to make the composite form normally without blocking the nozzle or breaking halfway [3]. Polyamide 66 (PA66) is a macromolecule containing amide groups in the main chain repeat unit. It has high wear resistance and toughness, and good fluidity [4]. In this paper, carbon powder was added into a PA66 matrix, and the composite was made into filaments used in FDM. We aimed to find a proper carbon mass fraction to minimize thermal deformation with FDM, and then we ensured suitable composite-processing parameters with the above carbon mass fraction.

At present, there is much research progress in composites reinforced with carbon products. Lu et al. mixed tetra-[alpha-(p-amino) benzyloxyl] phthalocyanine zinc (II) (ZnPc) and multiwalled carbon nanotubes (MWCNTs) to prepare nanoarchitectonic composites mixed with ZnPc-fluorinated multiwalled carbon nanotubes (mixed-ZnPc-fMWCNT) and linked ZnPc-fluorinated multiwalled carbon nanotubes (linked-ZnPc-fMWCNT). Multiple experiments showed that both nanocomposites had excellent stability and recyclability, and that they are promising candidates as ecofriendly photocatalysts for the degradation of organic dyes in aqueous environments [5]. Wu et al. added carbon powder to a polymer; by observing microstructure changes in the indented area, they calculated residual stress distribution in a carbon fiber (CF) imbedded in a CF-reinforced polymer composite using the atomic-force-microscopy (AFM) indentation technique. The stress-induced formation of nanoscale defects in the CF and their transformation into fractures were directly characterized [6]. Yang et al, mixed carbon powder with epoxy, the distribution of interfacial stress and debonding evolution in T700/M-epoxy was simulated by finite element method (FEM), and the procedure was experimentally verified [7]. Adding the T700/M-epoxy increases composite mechanical strength. Kamarian et al. added MWCNTs to the epoxy and observed the effects on the coefficient of the composite thermal expansion [8]. Adding MWCNTs to the epoxy decreased the coefficient of the composite thermal expansion. Gao et al. proposed a new adaptive method on the basis of carbon-fiber-reinforced plastics (CFRP) to reduce the thermal deformation of ball screws and improve accuracy [9]. Karthicksundar et al. infused carbon nano-powder into the epoxy matrix by using an ultrasonic liquid processor for homogeneity and to remove impurities by a vacuum pump to increase the thermal stability of materials used in the leading edges of wing and tail sections in almost all aircraft [10]. Vinyas et al. evaluated the mechanical and thermal properties of 3D-printed PLA and its composites, which are reinforced with carbon fibers. PLA mechanical strength with carbon powder was increased and inclusion of nylon glass fibers to PLA resulted in an improved thermal stability and the degradation temperature [11]. Tan et al. used an ultrasonically assisted stirring–mixing process to fabricate epoxy composites filled with pristine carbon nanotubes (P-CNTs) and functionalized CNTs (M-CNTs) that had increased glass-transition temperature and high thermal stability [12]. Ebrahimi and Qaderi focused on the buckling characteristics of polymer composites reinforced with graphene platelets (GPLs) in a thermal environment resting on viscoelastic foundation. By rising GPL weight fraction, critical buckling temperature increases and also critical buckling temperature increases when Winkler coefficient enlarges. By rising length/width ratio critical buckling temperature is reduced [13]. Dorigato et al. developed multifunctional epoxy-short carbon-powder-reinforced composites suitable for thermal-energy storage technology for the first time. CF introduction positively was contributed to the material stiffness and strength [14]. Zhu et al. presented bending and free-vibration analyses of thin–moderately thick composite plates reinforced by single-walled carbon nanotubes using the finite-element method based on the first-order shear-deformation plate theory. It was concluded that for the bending analysis, the carbon nanotubes (CNT) volume fraction, the width-to-thickness ratio and the boundary condition significantly influence the bending deflection, while the effect of the CNT volume fraction on the central axial stress can be neglected. For the free-vibration analysis, it is found that both the CNT volume fraction and the width-to-thickness ratio have pronounced effect on the natural frequencies and vibration mode shapes of the carbon nanotube-reinforced composites (CNTRC) plate [15].

On the basis of research, this paper studies the proper dose of carbon added to PA66 to find the appropriate process parameters of the composite to decrease the thermal deformation. 

## 2. Materials and Methods

### 2.1. Materials

In this paper, the material consisted of PA66 and carbon powder. PA66 is Zytel@103HSL NC010, and carbon powder, which was 0.01 mm particles (±0.005 mm), was from Shenzhen Turing Technology Evolution Co. Ltd (Shenzhen, China). With different carbon contents, composite filaments are produced by a twin-screw extruder after high-temperature drying. By an extruding experiment, the more carbon content there is, the more difficult that forming is. This is because that carbon powder decreases composite fluidity and makes the composite more brittle. The proper mass fraction of the carbon powder was 20%. In this content, the material is normally formed without blocking the nozzle or the composite filament breaking.

### 2.2. Methods

#### 2.2.1. Material Model

There are two simulation parts with regard to composite and processing modeling, respectively. Composite-material modeling is built by Digimat-MF. Digimat is linear and nonlinear multiscale-material modeling software (Digimat 2017.0, MSC software, Los Angeles, CA, USA) that accurately predicts the nonlinear microscopic behavior of complex multiphase composites and structures (e.g., PA, PA66, ABS, and nano-reinforced composites), and the constitutive behavior of various materials. The material properties of raw materials are input to the software user interface. 

The composites’ performance prediction by Digimat-MF is including thermal–mechanical analysis and thermal analysis.

1. Mechanical analysis

(1) Linear Elastic Model

The material model of carbon powder is a linear elastic model. The stress-strain relationship of elastomers can be described by Hooke’s law shown as Equation (1):*σ = C : ε*(1)

*C* is Hooke’s operator, *σ* is stress, and *ε* is strain.

The distribution of carbon powder in composites has no fixed direction. The properties of the material are independent of the loading direction under consideration. At this point, the Hooke’ operator only needs to be represented by two engineering constants: Young’s modulus and Poisson’s ratio shown as Equations (2) and (3).
(2)G=E/2(1 + υ)
(3)K=E/3(1 − 2υ)

*E* is elasticity modulus, υ is Poisson’s ratio, *G* is shear modulus, and *K* is bulk modulus.

(2) Elastic-plastic Model

The material model of PA66 is elastic-plastic model. The elastic-plastic constitutive model provided by Digimat-MF is *J_2_* plastic model. And the plastic cumulative hardening stress solution model is Equation (4): (4)$$R(p)=k\cdot {p\;+\;R}_{\infty }\cdot {[1-e}^{-\mathrm{mp}}]$$

R(p) is hardening stress, p is cumulative plastic strain, $R_{\infty }$ is hardening modulus, *m* is the hardening exponent, *k* is the linear hardening exponent.

With temperature changing, the thermal strain occurred to the material besides elastic strain and plastic strain. Thermal strain is isotropic, which is the function of actual temperature *T*, reference temperature *T_ref_* and initial temperature *T_ini_* shown as Equation (5):(5)$$\varepsilon^{\mathrm{th}}\left(T)=\{\alpha (T\right)\cdot {[T-T}_{\mathrm{ref}}{]-\alpha (T}_{\mathrm{ini}})\cdot {[T}_{\mathrm{ini}}{-T}_{\mathrm{ref}}]\}1$$

α(T) is the coefficient of thermal expansion.

2. Thermal analysis

Digimat-MF only provides a model for thermal analysis: Fourier model. The formula is shown in Equation (6).
(6)$$\rho \cdot c\frac{\mathrm{dT}}{\mathrm{dt}}=-\mathrm{div}(q)+r$$

ρ, *c*, *T*, *t*, *q*, *r* represent density, specific heat, temperature, time, heat flow, and volume heating, respectively.

If only a single thermal conductivity is considered, according to the Fourier law, the heat flow is expressed as Equation (7).
(7)$${q=-k}^{\mathrm{th}}\cdot \nabla \left(T\right)$$

$k^{\mathrm{th}}$ is thermal conductivity matrix. When the material is isotropic, the matrix can be expressed as Equation (8).
(8)$$k^{\mathrm{th}}=k\cdot \left(\begin{array}{ccc} 1 & 0 & 0\\ 0 & 1 & 0\\ 0 & 0 & 1 \end{array}\right)$$

*k* is thermal conductivity.

In this study, carbon powder was seen as a linear elastic model. The material properties needed to be input, including elasticity modulus, Poisson’s ratio, and coefficient of linear expansion. Then, carbon-powder size and shape were input. PA66 was viewed as elastoplastic material which is shown in Figure 1.

The constitutive curve needed to be imported to build the material model of PA66 and carbon powder. For this purpose, the latent heat of the phase change was expressed by equivalent heat capacity, the specific heat of PA66 changing with the temperature needed to be input. The specific heat capacity and constitutive curve of the raw materials were measured by differential scanning calorimeter (DSC, HESON, Shanghai, China) and universal electronic testing machine (Yangzhou drei instrument equipment co. LTD, Yangzhou, China), respectively. The remaining physical-performance parameters were given by the manufacturer. Carbon content was changed from 0% to 20% to observe the numerical results of linear expansion coefficients and the constitutive curves that are mainly material elements to affect thermal deformation; parameters are shown in Table 1 and Table 2. Digimat, based on mean-field homogenization, homogenized the original material. Therefore, thermal composite deformation with proper mass fraction was obtained by simulation, which was later experimentally verified. The predicted composite model was the material model in the ANSYS thermal–mechanical coupling.

#### 2.2.2. FDM Numerical Processing

Thermal–mechanical coupling analysis during FDM processing was carried out with ANSYS. ANSYS is a large general-purpose finite element analysis (FEA) software developed by ANSYS corporation of America (15.0, USA). It is used to solve structural, fluid, electrical, electromagnetic and collision problems. The birth and death element are a function in ANSYS. To achieve the “element death” effect, the program does not actually remove “killed” elements. Instead, it deactivates them by multiplying their stiffness (or conductivity, or other analogous quantity) by a severe reduction factor (ESTIF). This factor is set to 1.0 × 10^−6^ by default, but can be given other values. Element loads associated with deactivated elements are zeroed out of the load vector, however, they still appear in element-load lists. Similarly, mass, damping, specific heat, and other such effects are set to zero for deactivated elements. The mass and energy of deactivated elements are not included in the summations over the model. An element’s strain is also set to zero as soon as that element is killed. In like manner, when elements are “born”, they are not actually added to the model; they are simply reactivated. All elements are created, including those to be born in later stages of analysis, while in PREP7. It is not allowed to create new elements in SOLUTION. To “add” an element, the performer first deactivates it, then reactivates it at the proper load step. When an element is reactivated, its stiffness, mass, element loads, etc. return to their full original values. Elements are reactivated with no record of strain history (or heat storage, etc.); that is, a reactivated element is generally strain-free. Initial strain defined as a real constant, however, is not be affected by birth and death operations.

At the beginning, every element is “dead”. If the position accumulated by material is activated, the physical performance parameters are back to normal. The function was used to imitate the process of accumulating material. Every element that needed to be activated was a 0.4 × 0.4 × 0.2 mm^3^ cuboid. The whole model was 8 × 2 × 2 mm^3^, which means that the total number of elements were 1000. And there are 1386 nodes. At the beginning, every element was “dead”, which means that the element parameters were multiplied by a value close to 0. The accumulated position with the material was activated. Composite-material modeling in ANSYS was from the Digimat numerical results. The type of elements used in thermal analysis is solid70. Then, that in mechanical analysis is changed into an equivalent structural element solid45. The model surfaces except underside had natural convection with the air. Environment temperature was 15 °C. The underside surface had heat conduction with hot bed. The temperature of hot bed was 70 °C, 80 °C, and 90 °C. According to the experiment, the polymer’s thermal convection coefficient under natural convection was around 72. In this paper, PA66’s thermal convection coefficient was 72 and the thermal conductivity was predicted results from Digimat. The heat source was nozzle whose temperature was from 240 °C to 260 °C. The underside surface was stuck to the hot bed, so it had no deformation. The displacement of underside is 0 in three directions (x, y, and z). The results of thermal analysis are as the loads in mechanical analysis in order of time. 

The thermal–mechanical coupling simulated the temperature change during FDM. Then, warp and thermal-deformation displacement occurred. 

The parameters analyzed by ANSYS included nozzle and hot-bed temperature, the ratio of extrusion speed to filling speed, and aspect ratio. The reason for choosing these parameters is as follows. Hot-bed and nozzle temperature directly determine specimen temperatures. Aspect ratio is defined as, assuming that the nozzle moving direction is in the x direction of a specimen, the size ratio of the x direction to the y direction of a specimen is the aspect ratio. The aspect ratio affects specimen-temperature distribution. The ratio of extrusion speed to filling speed is an important element impacting processing quality. A poor ratio leads to the discontinuity of the formed material, and if there are bubbles in the specimens, the heat-conducting property and mechanical strength are affected. The numerical model was obtained shown in Figure 2.

#### 2.2.3. FDM Experiments

Specimens (50 × 30 × 8 mm^3^) were produced by I3 Reprap. The material was extruded from the nozzle to form the final shape. Then, the warp needed to be measured. The difference ratios between the maximum and minimum of the heights on specimen surface diagonals to the lengths of diagonals were the warp. In Figure 3, black lines form the contour after specimen thermal deformation occurred, orange lines are contour lines of the numerical model before thermal deformation. Meanwhile, H is the difference between the maximum and minimum of the heights on specimen surface diagonals. L is the diagonal length. Warp (W) was calculated as in Equation (9).
*W = H/L × 100%*(9)
where D is the maximum of thermal-deformation displacement on the diagonal. The height and length of the specimen surface diagonal were measured by a 3D laser measuring microscope. The 3D microscope is LEXT OLS4100 which is produced by the Olympus Corporation (LEXT OLS4100, OLYMPUS, Japan). The measured warp model is shown as Figure 3 of revision. 

The aim of this article was to make a composite material used in FDM to decrease thermal deformation and then obtain the optimal processing parameters of this material.

## 3. Results and Discussion

This paper predicted the difference of physical properties with different carbon content through Digimat that provide a material model base built in ANSYS. Processing temperature changes during FDM is achieved by thermal–mechanical coupling analysis in ANSYS. In this section, specimen thermal deformation was measured in a simulation and experiment.

### 3.1. Material

Table 3 and Table 4 show the predicted physical properties of the composite material with different carbon contents by Digimat. The coefficient of thermal expansion (CTE) decreased, while density (ρ), modulus of elasticity (E), Poisson’s ratio (ν), heat-conductivity coefficient (λ), and specific heat capacity increased by adding carbon powder, which is good for reducing thermal deformation during processing.

Actual conditions were experimentally determined. Specimen warpage was not seen by the naked eye; otherwise, the processing had failed. The specimen is shown in Figure 4. 

Table 5 shows the warp of PA66 with different carbon-content workpieces; 3 specimens were produced of each carbon content for comparison. Warp 1 is the warp of the first specimen with PA66 or PA66/carbon powder, Warp 2 is the second specimen with PA66 or PA66/carbon powder, and Warp 3 is the third specimen with PA66 or PA66/carbon powder. Calculated from the table data, the average warp of the PA66-forming workpiece was 0.1953%, and the average warp of PA66 with 20% carbon powder was 0.0982%. Through calculation, we found that the warp of the molding workpiece decreased by 49.7% after adding 20% carbon powder.

Table 6 shows the thermal deformation of a workpiece with different carbon contents. Comparing the experiment results, the maximal thermal-deformation displacement (Maxd) of the PA66 workpiece was, on average, 90.46 μm. The PA66 with 20% carbon powder had the least thermal-deformation displacement average of 80.167 μm. After adding carbon powder, the maximal thermal deformation displacement of the specimen was reduced by 11.4%.

### 3.2. Optimal Processing Parameters

#### 3.2.1. Nozzle and Hot-Bed Temperature

The temperature gap between nozzle and environment temperature was too large (though the forming room was sealed, forming-room temperature was not higher than the hot-bed temperature), so hot-bed temperature was needed as a moderating medium to slowly reduce the cooling speed of the material temperature.

However, when nozzle and hot-bed temperature was high, colloid viscosity on the substrate decreased, so improper hot-bed and nozzle temperature aggravated the warp of the formed workpieces.

In Figure 5, curves with different colors are different hot-bed temperatures, dotted curves are numerical results, and solid curves are experiment results. Both had the same trend in processing warp. The warp of the forming workpiece was the minimum when the temperature of the hot bed was 90 °C at a nozzle temperature of 240–250 °C. After a nozzle temperature of 255 °C, the warp of the forming workpiece was the minimum when the hot-bed temperature was 80 °C. The lower the nozzle temperature was, the more beneficial it was to reduce the warp. 

In Figure 6, curves with different colors are different hot-bed temperatures, dotted curves are simulation results, and solid curves are actual product results. Both had the same trend in processing warp. The lower the nozzle temperature and the hot-bed temperature were, the smaller the maximum thermal deformation and displacement of the workpiece were. Therefore, when the nozzle temperature was 240 °C and hot-bed temperature was 90 °C, the maximal thermal-deformation displacement and warp of the forming workpieces were minimal.

#### 3.2.2. Speed Ratio of Extrusion to Filling

In the actual forming process, the relationship between nozzle extrusion speed and filling speed severely affected processing quality, such as surface and size quality. Thermal deformation was indirectly affected. The relationship of extrusion speed and filling speed is shown in Equation (10):(10)$$\frac{v_{j}}{v_{t}}\in {(\alpha }_{1}{,\alpha }_{2})$$
where α_1_ is the critical value of the speed when there is a broken wire during forming, and α_2_ is the critical value of the block during forming; $v_{j}$ is extrusion speed; and $v_{t}$ is filling speed. If the ratio of extrusion speed to filling speed is *m* to have the best FDM forming effect, the value of *m* was obtained as in Equation (11):(11)$$m=\frac{\alpha_{1}+\alpha_{2}}{2}$$

Figure 7 shows the range of extrusion speed corresponding to filling speed at different nozzle temperatures. The abscissa is the nozzle temperature. The higher the nozzle temperature was, the larger the range of extrusion speed corresponding to the filling speed was. 

According to Equation (3), the values of *m* at different nozzle temperatures were calculated; Figure 8 shows these values, with the ordinate being the nozzle temperatures. These values were around 0.87. Through fitting, we found that the *m* of PA66/CF was 0.87.

#### 3.2.3. Model Aspect Ratio

Model aspect ratio is the ratio of the length of a specimen along the move direction of the nozzle to the length of a specimen perpendicular to the move direction. As is shown in Figure 9, it was about the simulation and experiment warp of different model aspect ratios. When the model aspect ratio was 0.25, the warp of the formed part was the minimum. Under the conditions of the same model, the warp of model aspect ratio that is a constant >1 was larger than that of a < 1 ratio.

Figure 10 shows the maximal numerical and experiment thermal-deformation displacement of different model aspect ratios. When the ratio was 0.5–1.3, the maximal thermal-deformation displacement was less than that of other model aspect ratios. However, different model aspect ratios had little impact on the maximal thermal-deformation displacement of the formed parts. Under the same model, with regard to the maximal thermal-deformation displacement of the workpiece, its model aspect ratio <1 was bigger than the >1 counterpart, which was constant. While model aspect ratio had little impact on maximal thermal-deformation displacement, the best model aspect ratio was 0.25.

## 4. Conclusions

In this paper, we studied the influence of FDM forming-process parameters on the thermal-deformation degree from the perspectives of composite material forming on an FDM basis, and the following conclusions were drawn:

(1) By establishing the composite-material model through Digimat-MF, the predicted mechanical- and thermal-performance parameters of the composite material were obtained. Limited by actual forming conditions, carbon-powder content of 20% (mass fraction) effectively reduced forming thermal deformation. Through experiments, we found that the addition of carbon powder reduced the warp of the forming workpieces by 49% and maximal thermal-deformation displacement by 11.4%.

(2) By obtaining optimized PA66/CF forming-process parameters on the basis FDM, carbon content was 20%. When forming nozzle temperature was 240 °C and hot-bed temperature was 90 °C, the maximal thermal-deformation displacement and warp of the forming workpiece were the minimum. Moving speed was 30 mm/s, extrusion speed was 12.5 mm/s, and the optimal ratio of extrusion speed and filling speed was 0.87.

## Figures and Tables

**Figure 1 materials-13-00519-f001:**
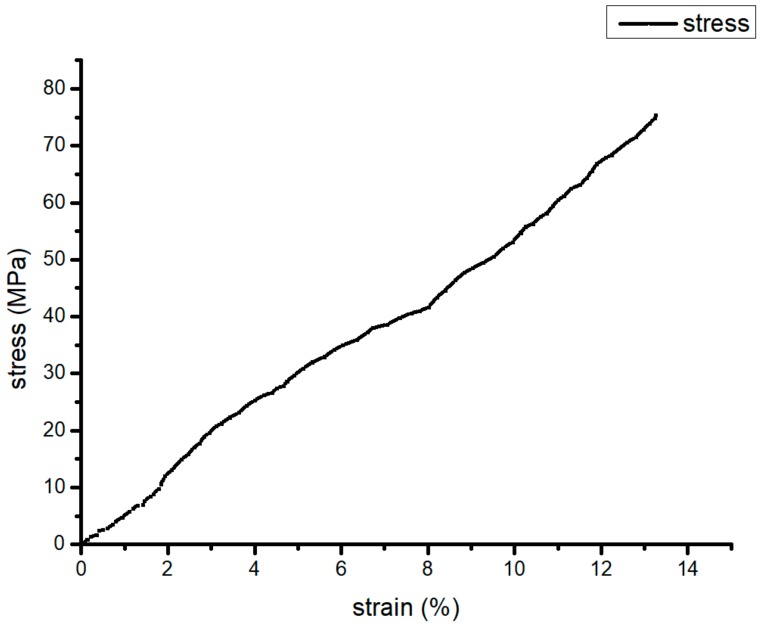
Elastoplastic curve of polyamide 66 (PA66).

**Figure 2 materials-13-00519-f002:**
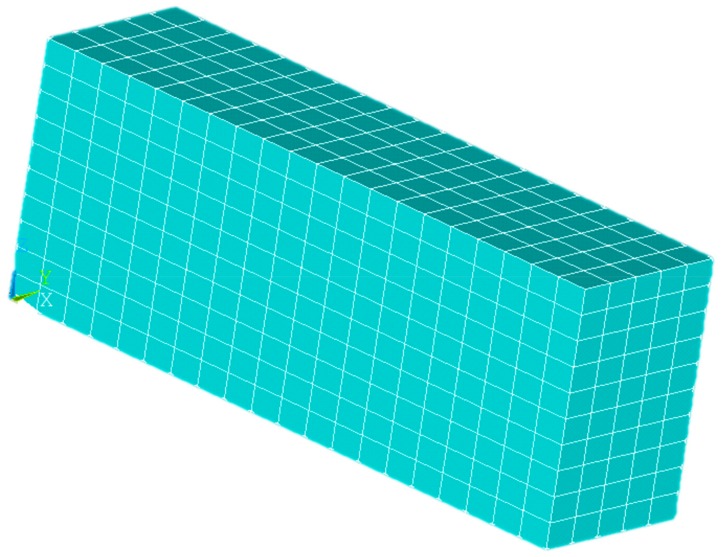
Numerical mesh in ANSYS.

**Figure 3 materials-13-00519-f003:**
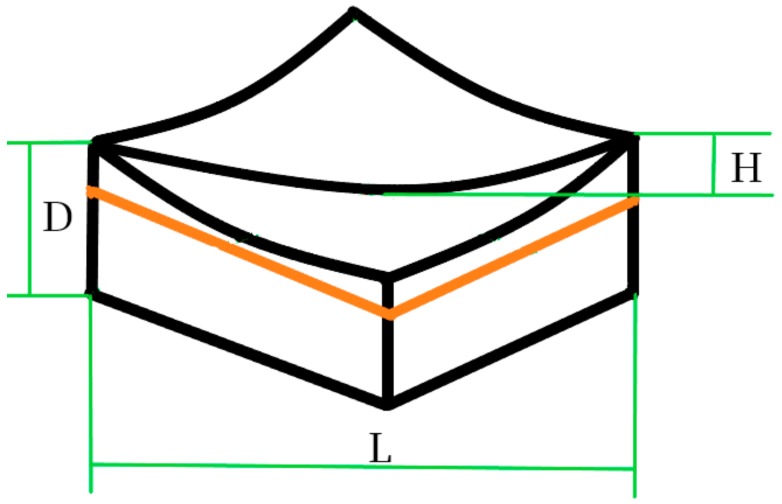
Warp model.

**Figure 4 materials-13-00519-f004:**
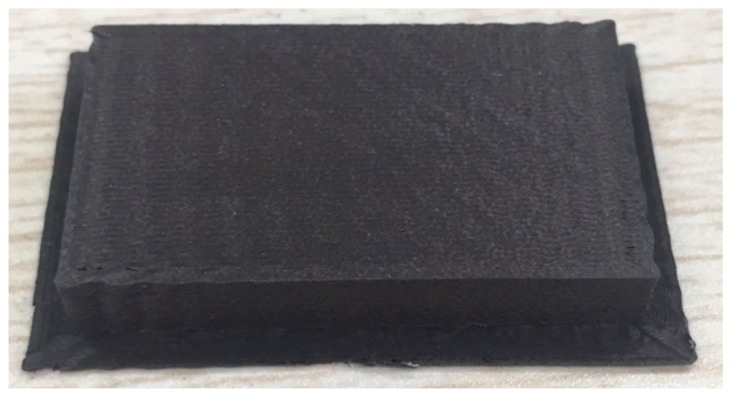
Composite specimen processed by fused deposition modeling (FDM).

**Figure 5 materials-13-00519-f005:**
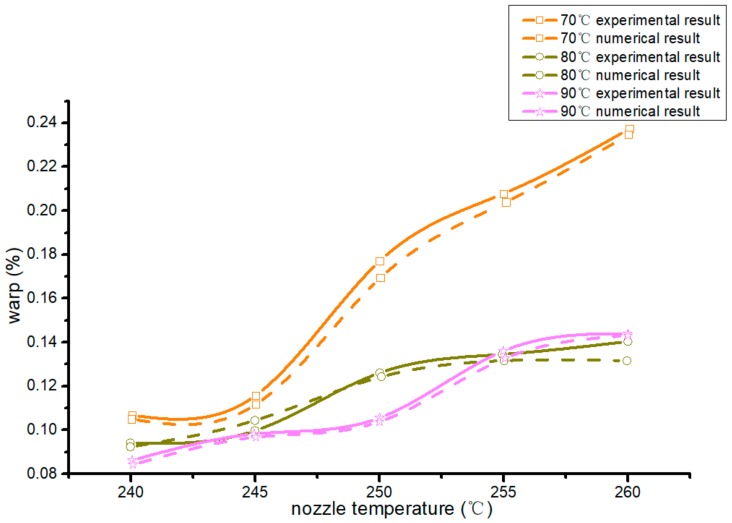
Numerical and experiment workpiece warpage at different hot-bed temperatures.

**Figure 6 materials-13-00519-f006:**
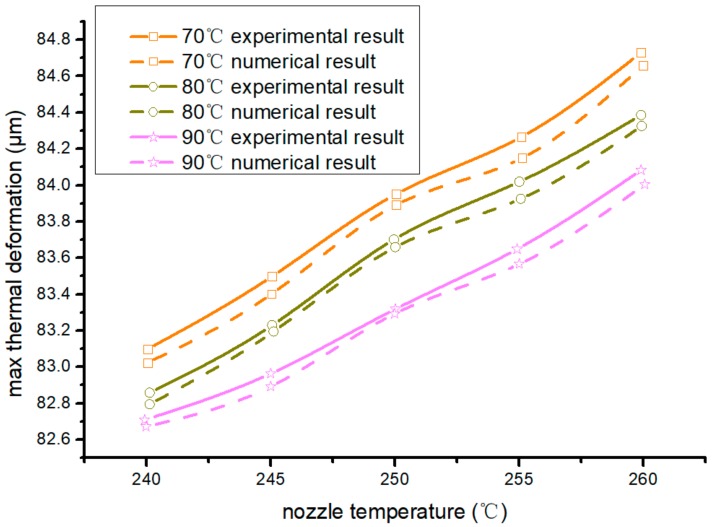
Maximal workpiece thermal-deformation displacement at different hot-bed temperatures.

**Figure 7 materials-13-00519-f007:**
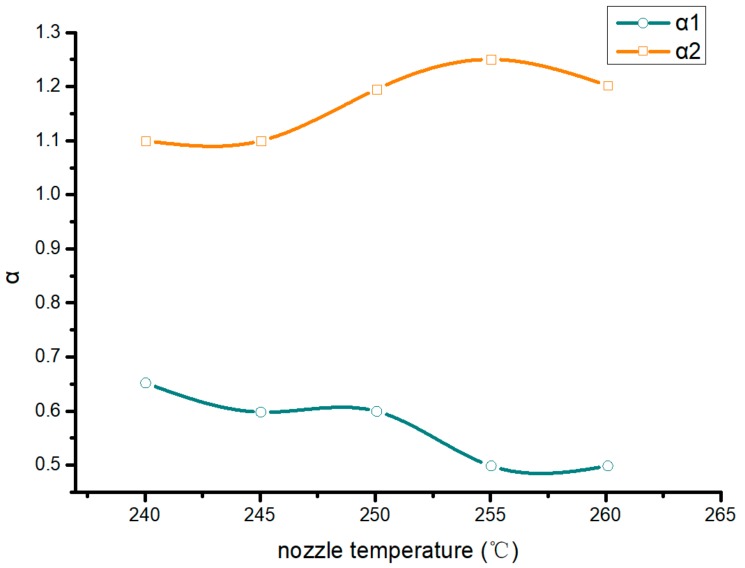
The ratios of extrusion velocity to filling velocity at different temperatures.

**Figure 8 materials-13-00519-f008:**
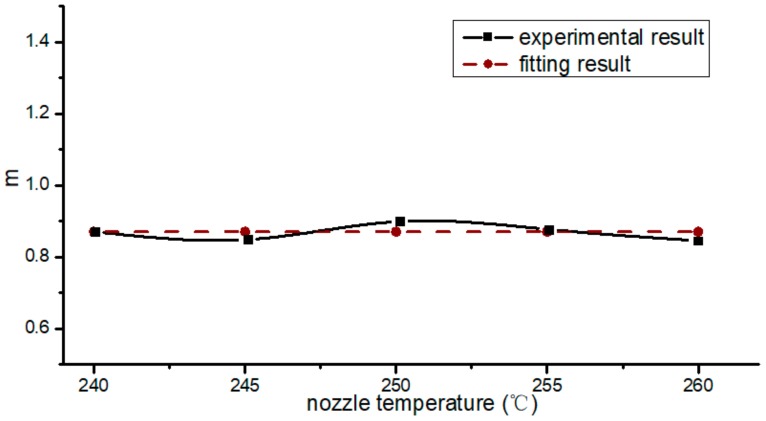
Optimal ratio of extrusion velocity to filling velocity at different temperatures.

**Figure 9 materials-13-00519-f009:**
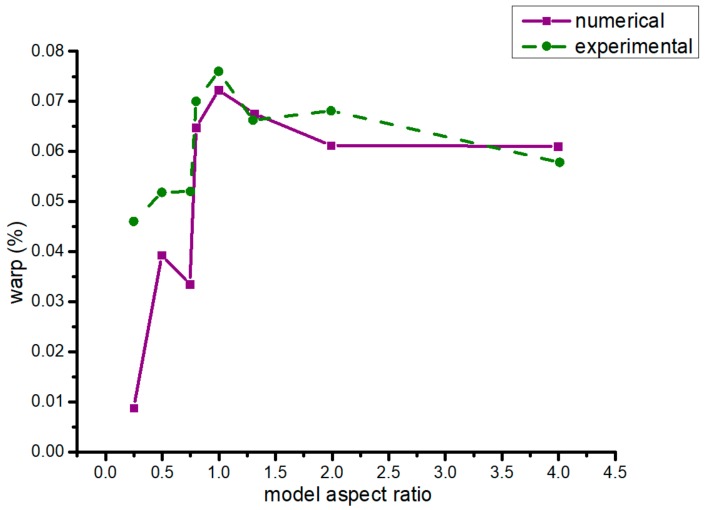
Warp in different model aspect ratios (numerical and experiment results).

**Figure 10 materials-13-00519-f010:**
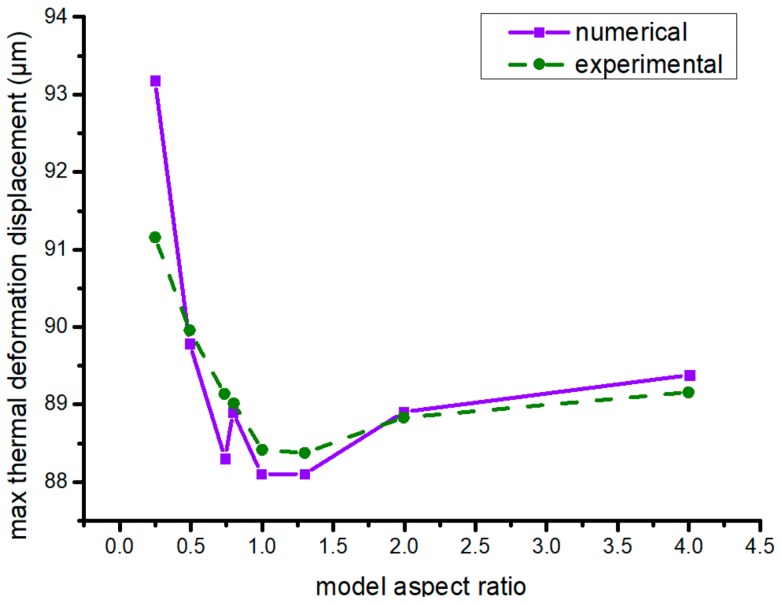
Maximal thermal-deformation displacement in different model aspect ratios.

**Table 1 materials-13-00519-t001:** Physical properties of raw materials.

Material	Unit	PA66	Carbon Powder
Density	$\mathrm{kg}/m^{3}$	1150	1750
Thermal conductivity	W/(m·°C)	0.35	10
Thermal expansion coefficient (×10^−5^)	m/°C	7.2	–3.8
Elastic modulus (×10^6^)	Pa	1200	228,000
Poisson’s ratio		0.4	0.307

**Table 2 materials-13-00519-t002:** Specific heat at different temperatures of PA66 and carbon powder.

Temperature (°C)	PA66’s Specific Heat Capacity (J/(kg·K))	Carbon’s Specific Heat Capacity (J/(kg·K))
15	681	7531
50	1849	7531
75	2029	7531
100	2171	7531
125	2468	7531
150	2795	7531
175	3186	7531
191	4499	7531
200	2595	7531
220	2429	7531
240	2412	7531
260	2428	7531

**Table 3 materials-13-00519-t003:** Numerical physical-property composite parameters with different carbon contents.

wt %	CTE (×10^−5^)	E (10^9^)	ν	ρ	λ
	1/°C	Pa		$\mathrm{kg}/m^{3}$	W/m·°C
20	5.8892	1.7632	0.3879	1234.7	0.508
15	6.2244	1.6275	0.391	1212.3	0.46
10	6.5545	1.5065	0.394	1190.8	0.42
5	6.8797	1.39702	0.397	1170.1	0.38

**Table 4 materials-13-00519-t004:** Specific heat capacity of composite materials with different carbon contents.

	Temperature (°C)	15	50	75	100	125	150	175	191	200	220	240	260
wt %													
5		1518	3708	4014	4189	4648	5259	6570	7815	6068	4670	4582	4626
10		1835	3868	4220	4448	4862	5358	6724	7842	6207	4820	4758	4794
15		2151	4052	4345	4580	4971	5440	6632	7864	6261	4971	4932	4946
20		2467	4275	4533	4790	5195	5618	6888	7886	6372	5140	5125	5085

**Table 5 materials-13-00519-t005:** Workpiece warp with different carbon contents.

wt %	Warp 1 (%)	Warp 2 (%)	Warp 3 (%)	Mean (%)
0	0.2013	0.1934	0.1911	0.1953
20	0.1028	0.0925	0.0994	0.0982

**Table 6 materials-13-00519-t006:** Maximal displacement of workpiece thermal deformation with different carbon contents.

wt %	Maxd1 (μm)	Maxd2 (μm)	Maxd3 (μm)	Mean (μm)
0	90.77	90.98	89.63	90.46
20	80.54	80.30	79.66	80.17

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
