# Peer review of "Thermal Deformation of PA66/Carbon Powder Composite Made with Fused Deposition Modeling"

_materials, 2020, doi:10.3390/ma13030519_

Round 1
Reviewer 1 Report
The paper fits well within the scope of this Journal. The authors after resubmission have improved the paper, however, major revisions are due.
The authors did not answer to the reviewer comment:
“The authors state to use the Digimat, but in my opinion the authors should clarify how they use it. Moreover, the state that they use Digimat with ANSYS, therefore they also should clarify this point.
They did not improve information about the model used in Ansys”. The authors should give more information about the model, mesh, boundary conditions, material model used in Ansys.
Table 1 the unit of measure for CTE is 1/°C
The authors should format the text in according to author guide of journal.
In the text there are several times the sentence “Error! Reference source not found.” Please delete them.
Figures 3 and 4: there are six curves in the figures, but the legend reports only three curves. Please correct it.
Figure 6: there are two curves in the figure, the authors should report in the figure the legend. The quality and resolution of the figure are low increase them.
Figures 7 and 8: change in the legend “simulation” with “numerical” and “experiment” with “experimental”.
Author Response
Dear editor and reviewers,
Thank you very much for your thorough reading of our manuscript and your valuable suggestions on improving our paper.
The corrections are listed below. Our specific responses with regard to the mandatory corrections are highlighted by red color in the manuscript.
Reviewer 1:
The English language is editing by MDPI editing service. “The authors state to use the Digimat, but in my opinion the authors should clarify how they use it. Moreover, the state that they use Digimat with ANSYS, therefore they also should clarify this point.We are grateful for the reviewer’s comment. Digimat is used to predict physical performance parameters of composite, and then the obtained physical performance parameters can be used to build a material model to complete the numerical analysis. The interface is shown in figure 1.
Figure 1. The guide of user interface
The MF module is mainly used. There are two kind of raw materials including PA66 and carbon powder. The PA66 is seen as elastic-plastic material whose parameters are as figure 2 and the constitutive curve is built as shown in figure 3. The carbon powder is seen as a linear elastic material whose parameters are as figure 4 and the constitutive curve is built as shown in figure 5. It is noted that the constitutive curve of PA66 is a curve with small radian which looks like a straight line. The abscissa axes of figure 3 and figure 5 is strain under uniaxial stretching and the longitudinal axes is axial force. Due to the phase transition occurring during processing, the specific heat capacity of PA66 mutates. Figure 6 is the change of specific heat capacity with temperature of PA66. The longitudinal axis is specific heat capacity of PA66. The phase of carbon is stable so the specific heat capacity is a constant.
Figure 2. physical performance parameters of carbon
Figure 3. The constitutive curve of carbon
Figure 4. physical performance parameters of PA66
Figure 5. The constitutive curve of carbon
Figure 6. The change of specific heat capacity with temperature of PA66
And then the physical performance parameters of composite are obtained. The material model of composite predicted by Digimat-MF is input into ANSYS in form of ADPL. The using process of Digimat is mentioned without elaborated exhaustively in manuscript because it would be a little basic and simple. If needed, the part could be added into the manuscript.
They did not improve information about the model used in Ansys”. The authors should give more information about the model, mesh, boundary conditions, material model used in Ansys.Thanks for the reviewer’s comments concerning our manuscript, which is of great significance to our research. The model is a 8×2×2 cuboid meshed into 1000 units. (Figure 7)
Figure 7. Numerical mesh in ANSYS
The surfaces except underside have natural convection with air. The environment temperature is 15℃. The underside surface has heat conduction with hot bed. The temperature of hot bed is 70℃, 80℃ and 90℃. According to the experiment, the polymer`s thermal convection coefficient under natural convection is around 72. In this paper, he thermal convection coefficient is 72 and the thermal conductivity is from predicted results from Digimat. The heat source is nozzle whose temperature is 240℃-260℃. The underside surface is stuck to the hot bed, so it is seen as no deformation. The material model is seen from table 1 from question 3.
The thermal-mechanical coupling simulates the temperature changing during FDM. And then the warp and thermal deformation displacement occur. This part has been modified which could be seen in chapter 2.2.2.
Table 1 the unit of measure for CTE is 1/°CThank you for this constructive suggestion. The unit has been corrected. (Table 2 in manuscript)
Table 1. Numerical physical property parameters of composites with different carbon content
|
wt.% |
CTE(×10-5) |
E(109) |
ν |
ρ |
λ |
|
% |
1/℃ |
Pa |
|
kg/ |
W/m·â„ƒ |
|
20 |
5.8892 |
1.7632 |
0.3879 |
1234.7 |
0.508 |
|
15 |
6.2244 |
1.6275 |
0.391 |
1212.3 |
0.46 |
|
10 |
6.5545 |
1.5065 |
0.394 |
1190.8 |
0.42 |
|
5 |
6.8797 |
1.39702 |
0.397 |
1170.1 |
0.38 |
Thanks for your very useful suggestion. The format complies with the template given by the official website of Materials strictly.
In the text there are several times the sentence “Error! Reference source not found.” Please delete them.We are grateful for the reviewer’s comment. The reference error has been deleted and the paper requotes correctly.
Figures 3 and 4: there are six curves in the figures, but the legend reports only three curves. Please correct it.Thanks for the reviewer’s comments concerning our manuscript. The point has been corrected. The dotted curves are the numerical results and the solid curves are experimental results. For example figure 8 and figure 9: (Figure 4 and Figure 5 in manuscript)
Figure 8. Numerical and experimental warp of workpiece
Figure 9. Maximum thermal deformation displacement of workpiece at different hot bed temperatures
Figure 6: there are two curves in the figure, the authors should report in the figure the legend. The quality and resolution of the figure are low increase them.Thanks for the reviewer’s comments concerning our manuscript, which is of great significance to our research. The figure has been modified shown in figure 10. (Figure 7 in manuscript)
Figure 10. The optimal ratio of extrusion velocity to filling velocity at different temperatures
Figures 7 and 8: change in the legend “simulation” with “numerical” and “experiment” with “experimental”.We are grateful for the reviewer’s comment. The “simulation” in the legend has been changed into “numerical” and “experiment” into “experimental” shown in figure 11 and figure 12. (Figure 8 and Figure 9 in manuscript)
Figure 11. The warp in different model aspect ratios (numerical and experimental result)
Figure 12. The thermal deformation displacement maximum in different model aspect ratios
Best regards,
Jingyu Sun.

Reviewer 2 Report
This paper is still very confusing. Experimental and modelling methods are poorly explained. English grammar is very poor. Please involve an expert in English language in order to improve the paper.
- Title: Change to “Thermal Deformation of PA66/ Carbon Powder Composite Made with Fused Deposition Modeling”
- Shouldn’t you use capital letters for your institution: “Department of Mechanical Engineering and Automation”?
- Do not use semicolons (;). Please shorten sentences.
- Do not use “it was known” or “it could be known” when presenting results. Instead use “it was found” or similar.
- Figures should be self-explaining. Describe within the figure what is experimental and simulation data.
- Use present instead of past tense. Exception: experimental test could be explained in past tense, since they were performed earlier.
- Please explain abbreviations at their first appearance, e.g. AFM, FEM. Do not repeat afterwards, e.g. multi-walled carbon nanotubes (MWCNTs) on page 1 and 2.
- Two versions of PLA on page 1 and 2: “polylactic acid” and “Poly (lactic-acid)”. Use only one. Afterwards stick to “PLA”.
- Do not cite first names, e.g. use “Wu et al.” instead of “Wu, MJ et al.”
- Literature review: Give results of previous studies. What was the effect of adding additional carbon material to resin?
- The study of Cheng et al. [12] on C/SiC does not fit into your topic. Please remove.
- I still do not understand how the composite specimens (e.g. the one shown in Figure 1) were produced. Neither do I understand how your model works. The description of the modelling is poor or missing at all. Experimental methods are not adequately described.
Since no line numbers are given, I can only provide page numbers when pointing out to specific text passages.
- Page 1: “It was restricted to be applied in broad fields…”. “Restricted” and “broad” are opposite things. Please re-write.
- Page 1: What does the “f” in “fMWCNT” refer to?
- Page 2: “It was obtained that mechanical strength and thermal properties of PLA with carbon fibers were increased [11].” What does it mean: thermal properties were increased? Increased coefficient of thermal expansion?
- Page 2: Last part of Section 1 (Introduction) is on results of the current study. Remove that part.
- Page 3, Section 2.2: What was actually simulated in Digimat and Ansys? The models are not given. Input parameters are provided, but where do they come from? What is the “life and death unit”?
- Page 3: “The material properties need to be input were elasticity modulus, Poisson's ratio, coefficient of linear expansion and so on.” What is “and so on”? Any other parameters?
- Page 3: Do not use “*”. Use “x” intead.
- Page 4: Section 3.1 is results, although is called “3.1 Mateial”. The first 2 sentences “Digimat is a linear…” and “The constitutive relation curve…” do not have any meaning for the results section.
- Table 1: Wher do parameters come from? Experimental tests or simulation results?
- Figure 2 A scale is missing (is 1 mm, 1 cm, 1m shown??). What do pictures show other then different grey values?
- Tables 2 and 3: What are 1, 2, and 3? Please provide equation how Warp and Maxd were calculated. Also a sketch of the specimen could help.
- Page 5: First part of 3.2.1. Nozzzle and hot bed temperature is redundant, since this information was provided before.
- Page 5: “The warp of forming parts increased with the rise of nozzle temperature. …”. This part refers to results. But where are these results presented? Is there a figure or a table?
- Results section: Please do not describe in the text what is presented in the axes of the diagrams. That could be seen from the figures! It is enough to write, e.g. “Figure 4 presents the average maximum thermal deformation as a function of nozzle temperature at different hot bed temperatures.”
- Equation 2: Parameter m is explained twice. Once is enough.
- Figure 5: What is the difference between the upper and the lower curves?
- Page 7: “The values of m at different nozzle temperatures were calculated from figure 5.” How were values of m calculated? What method was used?
- Page 8: “Under the condition of the same model, the warp of the model aspect ratio >1 was bigger than that <1.” Please add that it is constant >1.
Reviewer 3 Report
The authors did their best and corrected the work so now it can be published.
There are a couple of typos (3.1. The mateial, the progress had been achieved was impressive – maybe - that has been achived....., some sentences in chapter 2.2, etc .....)
I ask the authors to check the proper writing of the literature; is it enough to write only Y.S. or did you still need to put in a full last name from the author?
And there are a few more places left in the work where authors wrote carbon fiber. It should be replaced with carbon powder.
Round 2
Reviewer 1 Report
The authors have been improved the paper however major revision are still due.
1) The authors should report in table 2 also the values of specific heat of carbon for all temperatures.
2) The authors should show the elastoplastic curve of PA66.
3) The authors should clarify which model of material is used in the FEM analyses.
4) The authors should give more information about the model used in ANSYS: numbers of nodes, type of elements used in the thermo-mechanical analysis. Which are the boundary conditions in mechanical analysis?
5) The authors used the “element birth and death” technique they should give more information about it.
6) There are still some typos in the text. Please check it.
Reviewer 2 Report
The authors have significantly improved the paper. Some minor issues remain:
- No space between number and °C, e.g. 260°C.
- Line 51: “Yan” should be “Yang et al.” (reference [11]).
- Figure 1: Curve should start at [0;0].
- Table 2: Specific heat capacity should be given in J/(kg K), i.e. Kelvin instead of °C. Numbers remain the same.
- Section 2.2.2: I guess “unit” should be “element”. You are talking about finite elements, right?
- Line 142: 3 should be superscript in mm3.
- Figure 2: “mesh” instead of “meshes”.
- Line 171: A 3 is missing after mm, i.e. mm3.
- Line 174: “occurred” instead of “occurring”.
- Line 191: “show” instead of “shows”.
- Table 4: Table header is confusing, i.e. wt% and Temperature (°C).
Round 3
Reviewer 1 Report
The paper can be accepted in present form.
Author Response
Dear reviewer,
Thank you for your admitting. And I will try my best to study.
Best regards,
Jingyu Sun
This manuscript is a resubmission of an earlier submission. The following is a list of the peer review reports and author responses from that submission.
Round 1
Reviewer 1 Report
The authors wrote and covered this topic very poorly. And especially Chapter 3. There is a lack of introduction text on the processing parameters of FDM before starting to describe the results in Chapters 3.1 to 3.6.
Another question is how is carbon fiber added to the base material in FDM? Is it made immediately with a machine that has the capability (for example Markforge) or a fiber is mixed into the material and after a filament is made of it and then used as such in FDM?
At the outset, it cannot be concluded whether all the results were obtained by simulation alone or whether an analysis of the actual products made by the FDM was made. The title itself mentions the molding - whether it is really about making a mold with composite and put in for example injection machine or produce test specimen with FDM. But what these test specimen look like?
Is the deformation measured in comparison with the CAD model? Or?! Etc ....
Unfortunately, this work in its current form is not for publication.
Here are some more of my comments:
1.) The title itself tells us that this is a mold made by the FDM process? But reading further, I'm not really sure that the mold was made. These may be test specimens made with composite (PA66 and carbon fiber).
But what does this specimen look like? Authors should put at least a picture so the reader can understand what it is.
The authors state the molding part in the text. How can molding be, if it is FDM? Is it perhaps mold produce with FDM and put in injection molding machine and than tsting of warpage?!
2.) All measurements were made by simulation, there were no actual measurements on the finished sample? If not, what is the test specimen looks like? What are its dimensions? What machine was it made on?
3.) The warpage were compared according to the CAD model? In which software?
4.) When an abbreviation is mentioned for the first time, the full name should be written in the text, eg ABS, CTE, etc.
5.) Missing chapter on FDM? Why are the parameters chosen by the author important?
6.) What kind of software is Digimat? To simulate what?
7.) All text must be justify aligned.
8.) In Chapter 2.1. it was written that a multiphase composite simulation was performed with Digimat and then PA and PA66 were written. PA and PA 66 reinforced with eg fibers and particles should be added here. Because now it looks like that PA66 itself is thought to be multiphase material.
9.) In Figure 2, when indicating which curve indicates something, the full name should be written (eg PA66 with 5% carbon fiber, then PA66 with 10% carbon fiber, etc.)
10.) What in Figure 2 is strain 11 compared to strain 11. What is 11?
11.) Placed the explanation in the text before the figures 2 and 3. So first the explanation, then the figures.
12.) Why is Figure 2 put two times? Some picture.
13.) In Figure 3, mark each curve as I indicated under 9.)
14.) In Table 1, for the first time, some symbols appear, so you must put the full name. What is v? What is the unit of this measure? Is it a specific volume?
15.) What is CTE? If it is coefficient of liner thermal expansion - Isn't unit is 1/K?
16.) What does the word acording when below table 1 mean?
17.) The whole page 4 is in mess. Maybe when creating a pdf.
18.) The data in Table 1 was obtained from the software?
19.) What represent Figure 2.2?
20.) The text on page 4 is completely unrelated to the previous page. It looks like the text has neither a head nor a tail.
21.) Equation 2 is written twice in the text.
22.) The symbol m should be written italic.
23.) How the data in Tables 2 and 3 are measured? Comparison with the CAD model? Because are they a comparison to a real test body or just a simulation of a nozzle path?
24.) z-axis lifting - isn't that maybe layer thickness?
25.) The text below Table 3 is totally unrelated to the previous text. And is that not a Poisson factor?
26.) What does .Model ratio. mean just before Table 4?
27.) In Chapter 3:
- missing introductory text about the processing parameters that affect the product properties in the FDM
- what does three group experiment mean? Are these three specimens on which warpage were measured?
- in each table the mean and standard deviation are missing
- how were all the results obtained? Simulation in Digimat or measured on finished test specimens?
- the authors have the sentence: Table 6 is the thermal deformation of the workpiece of different carbon composites. Comparing the experiment results of three groups. - which three groups? When only the 0% and 20% carbon fiber composites are mentioned. And Table 1 and Figures 2 and 3 were made with 5, 10, 15 and 20% carbon fibers, respectively.
This whole analysis should be extended to other percentages of fiber. Now authors mention that the warpage with the 20% reinforcement is reduced, ok this is generally known. But it would be interesting to test some more percentages and see relations.
- attach an image of the manufactured product with some warpages.
- in section 3.3. it is mentioned that there is a temperature gap between the nozzle temperature and the enviroment – for which machine is this true? Because industrial FDM printers have a heated full chamber, so this difference can be greatly reduced. Alos there are 3D printers that do not have a heated chamber, but the chamber is closed and the environmental impact is greatly reduced. So this sentence doesn't make much sense, without further clarification about the processing process and the machines.
- which temperature is on the x axis in Figures 5 and 6? Nozzle temperatures?
- are the facts in Chapter 3.3. from the authors or is it from some literature?
- size and unit must be separated by ctrl + shift + space, not ordinary space.
- as shown in Figure 7, the warp of the forming workpiece is the minimum when the temperature of the hot bed is 90 ℃ at nozzle temperatures between 240-250 ℃. Does this mean PA66 or carbon-reinforced PA66, or a third material? It is not stated anywhere.
- figures 7 and 8 are of poor quality.
- chapter 3.4. is totally unrelated to the rest of the text.
28.) In Chapters 3.1 to 3.6. all the explanations of the tables are awfully written. It should be summed up in better sentences. This also applies to the text in conclusion in (2).
29.) Why in sections 3.4 and 3.6 are separated into the second row: As shown in ....
30.) From where in the conclusion does the fact that specific heat capacity is increased when it is nowhere in the text?
31.) Where is in the text citation from reference 17?
Reviewer 2 Report
I do not recommend the Manuscript for publication in “Materials”, due to the following main reasons:
Poor organisation of the Manuscript and incomprehensible description of the results. The Authors do not clearly define the considered problem. Lack of novelty. It seems that all the results are obtained by the use of a software Digimat-MF, with the application of standard built-in procedures. No validation of the obtained results is provided. Many language and editorial errors.The presented study does not bring any new interesting insight and the results are not even compared to any other method or experiment. In my opinion the presented research is too incremental to be published in a journal with high impact factor.
Reviewer 3 Report
This paper considerably lacks scientific standards. Materials and methods are mostly not provided. Results are not accurately discussed. This papers requires a major revision.
The term „FDM“ is not explained. Is it “Fused Deposition Modeling”?
The title is misleading, since you analyse thermal deformation of the composite and not of the molding. Please change to “Analysis of the thermal deformation of CF/PA66 composite made with Fused Deposition Modeling” or similar. Digimat is just a software. Please remove from the title.
Abstract:
In the Abstract, the technology (FDM) how composites were produced is completely missing. Please add some information. The first sentence of the abstract is excessively long. Please divide into 2 or 3 sentences. In the Abstract there should be some information on results and conclusions. Please add.
Introduction:
What is “ABS”?
There is another long sentence that requires revision: “When the thermal deformation of workpiece made of composite …”
Carbon fibers have a length of 10 micro meters. Is this still s fiber? Otherwise they seems to be longer in the SEM images. Please provide more details how length was measured and what is the variation in length. I guess not all fibers are exactly 10.0 micro meter.
Materials:
Please provide detailed information on what kind of materials were used and what methods were followed to manufacture composites.
Methods:
The only “method” given in the text is the software used for modelling: Digimat. In a scientific paper there more details are required on the methods used for modelling. Please provide more detailed information on the model.
Figures 2 and 3, Table 1: Here results are presented. Neither materials are defined, nor what kind of composites were manufactured with what kind of method. What kind of modelling method was used in order to obtain those results? What is “CTE” in Table 1?
There is a figure on Page 4 that is not explained. Also some figures are repeated in the text. I guess there is a problem with pdf processing. I would recommend removing links to figures in MS word.
Page 4: The term “formula” should be “Equation”.
What are “warpage” and “Maxd”? How were those values measured? Where are paths 11, 12, 21, and 22? I couldn’t find any information on the samples used for the measurements. Please provide much more details, such that the reader can follow your work.
Results:
Page 6. What are “m values”. Please provide more information and discussion on Figures 5 and 6.
Page 7: What are “continuous short-path” “discontinuous long-path”?
Reviewer 4 Report
This is a very interesting work that fits well within the scope of this Journal. However, the quality of the paper is very low, and some parts of the paper are difficult to understand. Some aspects need to be addressed prior to publication of this article. The paper is rejected.
The authors should explain all acronyms (FDM, ABS, etc.)
The authors state to use the Digimat, but in my opinion the authors should clarify how they use it. Moreover, the state that they use Digimat with ANSYS, therefore they also should clarify this point.
Pag. 2 fig. 2: the author should report the unit of measure on the axes.
Pag. 2 fig. 3: the author should report the unit of measure on the axes.
Figs.2 and 3: the authors should improve the quality and resolution.
Pag. 3: the fig. 2 is repeated.
Pag. 3: the figs. 2 and 3 should place after the text where they are cited.
Pag. 3: there is typo “According when” please check it.
Pag. 4: The formatting of text is wrong.
Pag. 4: the caption of figure is missing.
Pag. 4: the authors should clarify the sentence at pag. 4. What are the paths (11), (12), (21) and (22)?
Fig. 5: there are two curves; the authors should explain in the legend their sense.
Fig. 6: Improve the quality and resolution the legend is small.
Fig. 6: there are two curves; the authors should explain in the legend their sense.
Figs. 7 and 8: Improve the quality and resolution.
Fig. 8: Change “Maxd” with “Max deformation”
Extensive proof of English is due.